# Comparison of the Effectiveness of Liposuction for Lower Limb versus Upper Limb Lymphedema

**DOI:** 10.3390/jcm12051727

**Published:** 2023-02-21

**Authors:** Shuhei Yoshida, Hirofumi Imai, Solji Roh, Toshiro Mese, Isao Koshima

**Affiliations:** The Plastic and Reconstructive Surgery, the International Center for Lymphedema, Hiroshima University Hospital, 1-2-3, Kasumi, Minami Ward, Hiroshima 734-8551, Japan

**Keywords:** lymphedema, lower extremity, upper extremity, liposuction

## Abstract

Objective: Liposuction is the most frequently performed debulking procedure in patients with lymphedema. However, it remains uncertain whether liposuction is equally effective for upper extremity lymphedema (UEL) and lower extremity lymphedema (LEL). In this study, we retrospectively compared the effectiveness of liposuction according to whether it was performed for LEL or UEL, and identified factors associated with outcomes. Materials and Methods: All patients had been treated at least once by lymphovenous anastomosis or vascularized lymphatic transplant before liposuction but without sufficient volume reduction. The patients were divided into an LEL group and a UEL group, and then subdivided further according to whether they completed their planned compression therapy into an LEL compliance group, an LEL non-compliance group, a UEL compliance group, and a UEL non-compliance group. The reduction rates in LEL (REL) and in UEL (REU) were compared between the groups. Results: In total, 28 patients with unilateral lymphedema were enrolled (LEL compliance group, *n* = 12; LEL non-compliance group, *n* = 6; UEL compliance group, *n* = 10; UEL non-compliance group, *n* = 0). The non-compliance rate was significantly higher in the LEL group than in the UEL group (*p* = 0.04). REU was significantly higher than REL (100.1 ± 37.3% vs. 59.3 ± 49.4%; *p* = 0.03); however, there was no significant difference between REL in the LEL compliance group (86 ± 31%) and REU in the UEL group (101 ± 37%) (*p* = 0.32). Conclusion: Liposuction seems to be more effective in UEL than in LEL, probably because the compression therapy required for management after liposuction is easier to implement for UEL. The lower pressure and smaller coverage area required for postoperative management after liposuction in the upper limb may explain why liposuction is more effective in UEL than in LEL.

## 1. Introduction

The main treatments for lymphedema are conservative, including, in particular, compression therapy, or surgery. The surgical procedures are further classified into physiological reconstruction, which includes lymphovenous anastomosis (LVA) and vascularized lymphatic transplant (VLT), or debulking surgery, such as liposuction or direct excision [1]. LVA is generally used when lymphatic function is still preserved, and VLT and liposuction in more advanced cases where lymphatic function is severely impaired [2,3]. Debulking surgery plays a particularly important role in decreasing the volume of the affected extremity in patients with lymphedema and extensive deposition of fibroadipose soft tissue [4,5,6]. Liposuction has been the most frequently performed debulking surgery since the late 1980s [7,8,9].

Rigorous compression therapy is required after liposuction for severe lymphedema as a routine part of postoperative care [9,10,11]. We have previously demonstrated the importance of effective compression therapy for the success of liposuction for lymphedema and described the compression procedure used after liposuction therapy in patients with lower extremity lymphedema (LEL) [12]. However, it has been unclear whether liposuction is as effective for upper extremity lymphedema (UEL) as it is for LEL. In this study, we compared the effectiveness of liposuction in our patients with LEL with that in those with UEL and sought to identify factors associated with the outcomes.

## 2. Methods and Patients

Patients who underwent liposuction for unilateral LEL or UEL at Hiroshima University Hospital between May 2019 and June 2021 were retrospectively identified. LVA and VLT had been performed at least once before liposuction in all patients, but volume reduction had been insufficient in all cases.

The study was approved by the Hiroshima University Hospital ethics committee (approval number: E-1413) and conducted in accordance with the Declaration of Helsinki and the STROBE guidelines (http://www.strobe-statement.org/ accessed on 1 April 2019.). All study participants provided written informed consent.

The exclusion criteria were as follows: body mass index (BMI) > 35; ambulation with support/auxiliary aids; comorbidity of cardiac failure, renal insufficiency, cirrhosis hepatis, hypalbuminemia, deep venous thrombosis, chronic venous insufficiency, thyroid dysfunction or other endocrine dysfunction, drug-induced peripheral edema, or arterial and/or venous malformation; and inability to implement compression therapy.

Preoperatively, JOBST^®^ stockings and sleeves (BSN-JOBST GmbH, Hamburg, Germany) were used to apply compression pressure of 20–30 mmHg on the lower leg, 10–15 mmHg on the thigh, 10–20 mmHg on the forearm, and 5–10 mmHg on the upper arm.

Indocyanine green (ICG) lymphography was performed in all patients before surgery. First, 0.2 mL of ICG (Diagnogreen^®^ 0.25%; Daiichi Sankyo, Tokyo, Japan) was injected subcutaneously into each affected extremity at the first web space of the foot or hand and at the lateral border of the Achilles tendon or the palmar surface on the ulnar side of the wrist. ICG lymphography images in the plateau phase (8–24 h after the injection) were obtained circumferentially using an infrared camera system (Photodynamic Eye; Hamamatsu Photonics K.K., Hamamatsu, Japan).

Lymphatic function in the affected limb was also assessed preoperatively by lymphoscintigraphy at 10, 60, and 120 min after injection. A small amount (0.2 mL, 40 MBq) of technetium-99m-labeled human serum albumin was injected subcutaneously at the first web space of the foot or hand and at the lateral border of the Achilles tendon or the palmar surface on the ulnar side of the wrist. A gamma camera was used to obtain images of the limb.

The inclusion criteria were as follows: no weight gain after starting treatment at our institution; a diffuse pattern seen throughout each limb on ICG lymphography; and type III lymphedema seen on lymphoscintigraphy [13] according to the Society of Lymphology (ISL) classification [14].

Tape measurements were obtained at five anatomic locations on each limb (10 cm above the knee or elbow, at the knee or elbow, and 10 cm below the knee or elbow, ankle or wrist, and foot or palm in the supine position).

The LEL or UEL index was calculated on both sides in each case using the following equation [15,16]:LEL or UEL index = [total of the squares of the circumference at the five locations on each extremity]/BMI.

The percentage of excess LEL index for the affected lower extremity (PEL) or the percentage of excess UEL index for the affected upper extremity (PEU) was calculated for each case as follows:PEL = ([LEL index of affected extremity] − [LEL index of the unaffected contralateral extremity])/[LEL index of the unaffected contralateral extremity] × 100
PEU = ([UEL index of affected extremity] − [UEL index of the unaffected contralateral extremity])/[UEL index of the unaffected contralateral extremity] × 100.

The LEL index, UEL index, PEL index, and PEU index were calculated for each case immediately before and 6 months after liposuction and at the end of the observation period.

The improvement rate in the LEL or UEL index was calculated for each case as follows: ([preoperative LEL index or UEL index] − [postoperative LEL index or UEL index])/[preoperative LEL index or UEL index])] × 100.

The reduction rate in PEL (REL) or PEU (REU) after liposuction was calculated for each case as follows:

REL or REU = ([preoperative PEL index or PEU index] − [postoperative PEL index or PEU index])/[preoperative PEL index or PEU index] × 100 (Table 1).

The rates of improvement in the LEL index, UEL index, REL index, and REU index were calculated at 6 months after liposuction and at the end of the observation period.

## 3. Surgical Procedures

### 3.1. Liposuction

Extensive liposuction was performed circumferentially via approximately 10 skin incisions each measuring 3–4 mm in length using cannulas with a diameter of 6 mm from the ankle to the hip in patients with LEL, and from the wrist to the shoulder in those with UEL using a non-assisted liposuction device (Lead S-200, Kakinuma Medical, Inc., Tokyo, Japan). Referencing the size of the unaffected extremity, as much subcutaneous fat and fibrotic tissue as possible was removed while sparing the LVA or VLT sites to avoid injury [17,18,19]. Using the tumescent technique combined with a tourniquet, 1000 mL of saline mixed with 0.2 mg of adrenaline and 20 mL of lidocaine were infused subcutaneously immediately before liposuction to minimize blood loss [20,21]. After completion of liposuction in the areas distal to the tourniquet, a cotton compression bandage was applied from the most peripheral area to the most proximal area to minimize hemorrhage. Liposuction in the most proximal area was performed under the tumescent technique after the tourniquet was removed. After the liposuction in the most proximal area was completed, the compression bandage was rolled up to the most proximal area of the limb and attached on the trunk using adhesive tape. In the event that the proximal area was too large for the tourniquet to stop bleeding successfully [20,22], liposuction was performed in the proximal area first under the tumescent technique before clamping the tourniquet to reduce the volume of the proximal area until the tourniquet became effective. The total volume of fat removed was calculated. The incisions were not sutured to allow drainage postoperatively.

### 3.2. Postoperative Management

A cotton compression bandage was applied continuously for 7 days after liposuction. The pressure was adjusted to ≥40 mmHg on the lower leg, ≥20 mmHg on the thigh, ≥20 mmHg on the forearm, and ≥10 mmHg on the upper arm.

Compression therapy was applied for 6 months postoperatively as follows: for LEL, a compression stocking (JOBST Bellavar^®^; BSN-JOBST Inc., Conover, NC, USA) and bandaging (JOBST Comprihaft^®^); for UEL, a compression sleeve and glove (JOBST Bella Strong arm sleeve, glove^®^) and bandaging (JOBST Comprihaft^®^) with the pressure adjusted to be the same as that immediately after liposuction. All patients were instructed to wear the compression garment all day long continuously for 6 months after discharge, except when bathing. Instructions on self-care, such as self-lymph drainage, were provided at the first visit to our outpatient clinic; however, additional rehabilitation, such as lymph drainage, was not performed except for the instruction about wearing the compression garments before and after liposuction.

### 3.3. Follow-Up

After discharge, follow-up examinations were conducted monthly until postoperative month 12- and at 3-month intervals thereafter. At each visit, the patients were examined for skin ulceration, garment slipping, and their compliance with application of compression pressure. The compression pressure applied was also measured. Compliance with application of compression pressure was assessed by observing how the compression garments were worn and interviewing the patients regarding their daily routine for applying them. If a patient reported that it was not possible to continue their planned compression therapy for the first 6 months after surgery another option, such as a single stocking or sleeve, Velcro-type compression, or a cushion under wrapping, was used. The average of the measured pressure during the first 6 months after surgery was calculated for each patient. After 6 months of compression, the applied pressure was decreased and adjusted to be ≥20 mmHg on the lower leg, ≥10 mmHg on the thigh, ≥10 mmHg on the forearm, and ≥5 mmHg on the upper arm. The compression pressure applied was measured at each subsequent visit, and the average between 7 months and the last month of observation was calculated for each patient. The measurement points of compression pressure were at the front midpoint on the lower leg or on the palmar side at the midpoint on the forearm, and on the thigh or upper arm. All pressure measurements were performed using a pressure sensor device (Palm Q^®^; Cape Co. Ltd., Yokosuka, Japan).

### 3.4. Group Classification

The patients were divided into an LEL group and a UEL group. Patients in each group were then subdivided further into those who completed 6 months of compression therapy (LEL compliance group and UEL compliance group) and those who did not (LEL non-compliance group and UEL non-compliance group).

### 3.5. Statistical Analysis

The data are described as the mean and standard deviation (range). The following variables were compared between the LEL and UEL groups using Student’s *t*-test: mean age, BMI, observation period, average time difference between the last physiological reconstruction (LVA or VLT) and liposuction, number of LVAs in the affected extremity, number of VLTs in the affected extremity, preoperative PEL or PEU index, total volume of liposuction, rate of improvement in LEL or UEL, the PEL or PEU index at the end of observation, REL or REU, mean compression pressure between 0.5 and 6 months, and mean compression between month 7 and the last month of observation. The sex distribution and implementation of the planned compression for the first 6 months after liposuction were compared between the LEL and UEL groups using the chi-squared test. The improvement rate in the LEL or UEL index, REL or REU, and mean compression pressure were compared between the compliance and non-compliance subgroups using the Tukey–Kramer test. The transitions of the LEL or UEL index and the PEL or PEU in the affected limb between before surgery and 6 months postoperatively through to the last month of observation were compared using the Tukey–Kramer test. All statistical analyses were performed using Statcel 4 software (OMS Publishing, Inc., Tokyo, Japan). A *p*-value < 0.05 was considered statistically significant.

## 4. Results

There were no complications during any of the surgical procedures. One patient developed a 5 × 3 cm skin ulcer postoperatively on the lateral side of the lower leg, which epithelialized after 2 months of conservative measures. There were no cases of cellulitis during the study period.

In total, 28 patients with unilateral lymphedema involving 18 lower extremities and 10 upper extremities were enrolled. There was no significant difference in the sex ratio (*p* = 0.9), distribution of patients with primary and secondary lymphedema (*p* = 0.9), age (*p* = 0.8), BMI (*p* = 0.06), observation period (*p* = 0.27), average time difference between the last physiological reconstruction (LVA or VLT) and liposuction (*p* = 0.67), number of LVAs (anastomoses) (*p* = 0.3) or VLTs (lymph node transfers) in the affected extremity (*p* = 0.7), or the preoperative PEL or PEU index (*p* = 0.6) between the LEL and UEL groups. The values for total liposuction volume (LEL; 2242 ± 1190 mL, UEL; 1260 ± 622 mL, *p* = 0.029), the postoperative PEL index vs. the PEU index at last month of the observation period (LEL; 14.9 ± 15.8%, UEL; 1.2 ± 13.1%, *p* = 0.03), and the mean compression pressure applied to the lower leg and thigh vs. the forearm and arm (initial 6 months after liposuction; foot to leg 34.4 ± 9.7 mmHg, hand to forearm 23.2 ± 2.0 mmHg, *p* = 0.001; thigh 17.5 ± 4.3 mmHg, arm 14 ± 1.7 mmHg, *p* = 0.02), (between 6 months and last observation after liposuction; foot to leg 23.4 ± 4.3, hand to forearm 12.1 ± 2.0 mmHg, *p* = 0.03 × 10^−6^; thigh 10.9 ± 1.1 mmHg, arm 6.8 ± 1.0 mmHg, *p* = 0.04 × 10^−8^) was significantly higher in the LEL group than in the UEL group throughout the observation period. The improvement rate in REU compared with REL was significantly higher in the UEL group than in the LEL group at last month of the observation period (LEL; 16.6 ± 9.3%, UEL; 24.3 ± 8.9%, *p* = 0.04). The rate of non-compliance during the initial 6 months of planned compression therapy after liposuction was significantly higher in the LEL group than in the UEL group (LEL; 6 cases, UEL; 0 cases, *p* = 0.04) (Table 2). There were no cases of non-compliance in the UEL group. In the LEL group, the reasons given for non-compliance during the initial 6 months of compression therapy after liposuction were pain (4 cases) and feeling too hot (2 cases). 

The transitions of the LEL index (pre ope; 298 ± 42, 6 months; 247 ± 33, last months; 248 ± 32), PEL (pre ope; 38 ± 21%, 6 months; 14 ± 17%, last months; 15 ± 16%), UEL index (pre ope; 124 ± 19, 6 months; 91 ± 11, last months; 93 ± 11), and PEU (pre ope; 34 ± 10%, 6 months; −0.5 ± 14%, last months; 1 ± 13%) were significantly lower at 6 months after liposuction and at the final observation than before liposuction (*p* < 0.01); however, there were no significant differences in the values for any of these variables between the assessment at 6 months and the final assessment after liposuction (*p* > 0.05) (Figure 1 and Figure 2).

The improvement rates of LEL and UEL (Figure 3) and of REL and REU (Figure 4) were significantly lower in the LEL non-compliance group (6 ± 6.7%) than in the LEL compliance group (22 ± 4.7%; *p* < 0.01) and UEL compliance group (24 ± 8.9%; *p* < 0.01). However, there was no significant difference between the LEL compliance group and the UEL compliance group in terms of the improvement rates of LEL and LEU (*p* > 0.05) (Figure 3). In addition, the REL or REU were significantly lower in the LEL non-compliance group (12 ± 42%) than that in the LEL compliance group (86 ± 31%; *p* < 0.01) and UEL compliance group (101 ± 37%; *p* < 0.01). However, there was no significant difference between the LEL compliance group and the UEL compliance group in the (*p* > 0.05) and of REL and REU (Figure 4).

Compression pressures on the leg or forearm were significantly higher in the LEL compliance group (43 ± 2.5 mmHg; *p* < 0.01) than that in the LEL non-compliance group (23 ± 9.8 mmHg; *p* < 0.01) and UEL compliance group (23 ± 2.0 mmHg). However, there was no significant difference between the LEL non-compliance group and the UEL compliance group (*p* > 0.05) (Figure 5, left panel). In addition, compression pressure on the thigh or upper arm were significantly higher in the LEL compliance group (22 ± 2.1 mmHg; *p* < 0.01) than in the LEL non-compliance (13 ± 4.2 mmHg) and UEL groups (14 ± 1.7 mmHg). However, there was no significant difference between the LEL non-compliance group and the UEL compliance group (*p* > 0.05) (Figure 5, right panel). Representative cases are shown in Figure 6, Figure 7 and Figure 8.

## 5. Discussion

In this study, the LEL and UEL groups were enrolled using the same inclusion and exclusion criteria, and there was no significant between-group difference in the baseline severity of lymphedema, as indicated by the preoperative PEL and PEU index values, the ISL classification, and findings on ICG lymphography or lymphoscintigraphy. Therefore, these two groups were considered suitable for comparison of the results of liposuction for LEL and UEL.

The total volume of liposuction was significantly higher in the LEL group; however, the improvement rate and REL and REU were significantly higher in the UEL group, indicating that liposuction is more effective for UEL.

The compression pressure applied after liposuction was significantly higher in the LEL compliance group than in the LEL non-compliance and UEL groups. Furthermore, the rate of improvement and REL and REU were significantly lower in the LEL non-compliance group than in the LEL compliance and UEL groups, with no significant difference between the LEL compliance group and the UEL group in terms of improvement in REL or REU. These findings indicate that a higher postoperative compression pressure is required in LEL than in UEL to obtain a good result after liposuction. The compression pressure required on both the peripheral and proximal portions of the extremity for postoperative management after liposuction was significantly higher in our LEL group than in our UEL group; the pain caused by this higher compression pressure may explain the greater number of patients in the LEL non-compliance group.

Our findings suggest that liposuction is more effective for UEL than for LEL because the compression therapy required after the procedure is easier to implement in the upper extremity. Another reason why compression therapy is more difficult after liposuction for LEL may be that some patients feel too hot when such a large area is covered by a thick compression garment.

During orthostasis, gravity causes a greater increase in microvascular filtration pressure in the lower limbs than in the upper limbs; therefore, more lymph accumulates at a lower level than at a higher level in the body [23]. This mechanism explains why the lower limbs need greater compression during orthostasis and why the compression pressure needs to be higher in patients with LEL than in those with UEL. However, a higher compression pressure is more burdensome for patients and more difficult to sustain.

Our previous findings highlighted the need to balance ease of wear, stability, and high pressure in the postoperative management of patients who have undergone liposuction for lymphedema [12]. Multilayer bandaging combined with a compression stocking is an effective way of accomplishing this balance [24,25,26,27,28]. Therefore, multilayer compression bandaging over a compression stocking was applied in our study.

According to Laplace’s law, when the same pressure is applied at different points on the surface of an extremity, the pressure inside the extremity is higher where the limb circumference is narrower [29,30]. Therefore, subcutaneous soft tissue pressure is not always reflected by surface pressure. Moreover, in theory, the pressure in the subcutaneous soft tissue decreases with increasing depth from the surface. This phenomenon becomes more pronounced as the circumference of the limb increases but is latent in limbs with a smaller circumference. Furthermore, according to the same theory, the pressure in the underlying subcutaneous soft tissues decreases in a manner that is inversely proportional to the circumference of the limb [22]. Therefore, compression therapy is thought to be more effective at the same pressure for lymphedematous extremities with less soft tissue after liposuction. The effect of liposuction on limbs with lymphedema may be explained by this theory.

Our study findings also suggest that it may be possible to decrease the compression pressure over time. Two mechanisms are thought to be potentially involved. In our study, LVA and VLT were performed at least once in advance of liposuction in all patients. In general, patients with reversible lymphedema (ISL stage I, mild) benefit most from a physiological reconstruction procedure, which can reduce the risk of progression to a more chronic advanced stage. It is recommended that reduction procedures be reserved for patients with severe advanced lymphedema (ISL stage II and particularly stage III) [31,32,33,34,35]. The first potential mechanism is that even patients with severe advanced lymphedema can benefit from a physiological reconstruction at least to the extent that it is possible to decrease the compression pressure after liposuction [18,35,36,37], even though there may not be sufficient improvement in terms of volume reduction after LVA or VLT.

Lymphangiogenesis is thought to potentially be involved in the second mechanism. Vascular endothelial growth factor (VEGF)-C, which promotes lymphangiogenesis [38,39,40,41], has been found to be overexpressed in lymphedematous limbs [42,43]. Although it is not clear whether liposuction damages the lymphatics [44,45], overexpression of VEGF-C may promote lymphangiogenesis after liposuction and make it possible to decrease the compression pressure after postoperative month 6. However, overexpressed VEGF-C has also been found to aggravate lymphedema by promoting vascular leakage [42,43]. These conflicting two features of VEGF-C, namely, its ability to promote lymphangiogenesis and to aggravate lymphedema by promoting vascular leakage, may be the reason why a high compression pressure is necessary for the initial 6 months of management after liposuction and why it is possible to decrease the compression pressure after a further period of time.

Given that volume reduction was prioritized in our study, a 6 mm cannula was used for liposuction; however, a 3 mm cannula is recommended from the viewpoint of cosmesis.

This study has some limitations. First, it involved a small number of patients, and the longest observation period was 25 months. Therefore, longer-term investigations in a larger sample are necessary. There is also a need for a more accurate assessment method. For example, we assessed compliance with application of compression pressure only by observation and interview at each follow-up visit during the study. A more objective method, such as having patients record daily compliance with application of compression pressure, may produce more accurate results.

## 6. Conclusions

Our results suggest that liposuction is more effective for patients with UEL than for those with LEL who have previously undergone LVA and VLT because the compression therapy required after liposuction is easier to apply in the upper limb than in the lower limb. A possible explanation for this finding is the lower pressure required and the smaller area needing to be covered after liposuction in the upper limb.

## Figures and Tables

**Figure 1 jcm-12-01727-f001:**
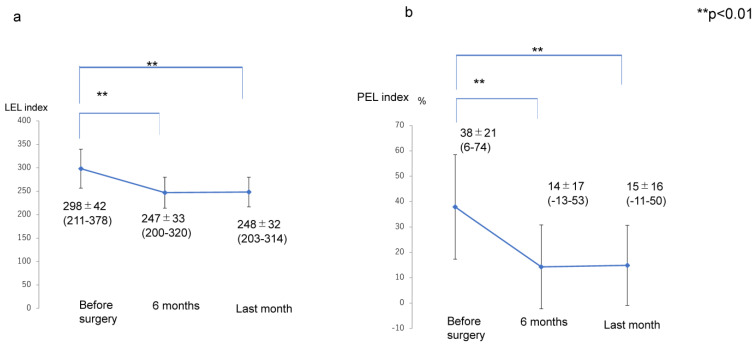
Transition of the LEL index and PEL index after liposuction for lymphedema in the affected limb. (**a**) LEL index. (**b**) PEL index. Abbreviations: LEL index, lower extremity lymphedema index; PEL, percentage of excess LEL index in the affected extremity.

**Figure 2 jcm-12-01727-f002:**
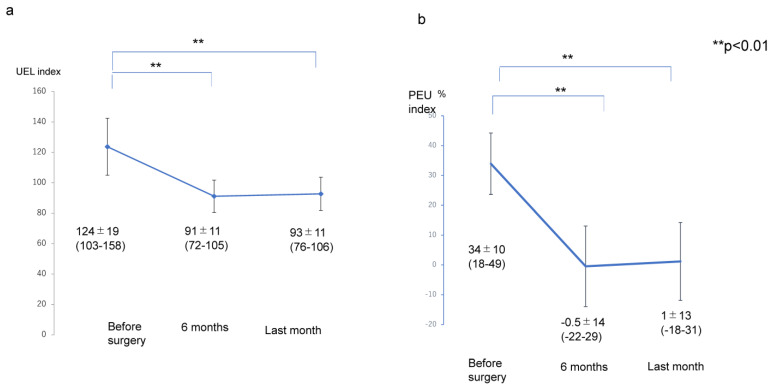
Transition of the UEL index and PEU in the affected extremity. (**a**) Transition of UEL index. (**b**) Transition of PEU. Abbreviations: PEU, percentage of excess UEL index in the affected extremity; UEL index, upper extremity lymphedema index.

**Figure 3 jcm-12-01727-f003:**
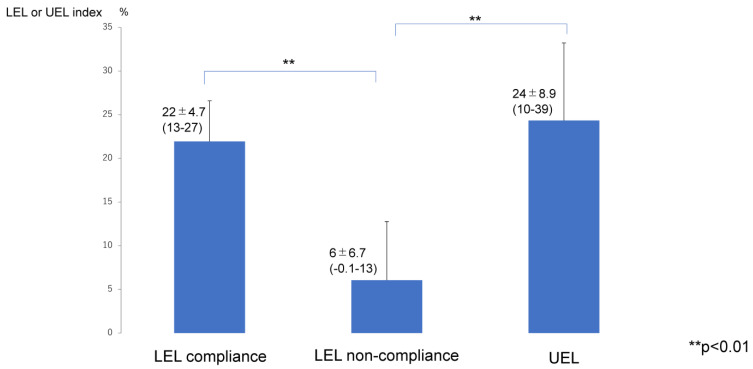
Rate of improvement in LEL or UEL after liposuction. LEL compliance group, patients who completed the planned compression therapy for the first 6 months after liposuction for lymphedema in the lower limb. LEL non-compliance group, patients who did not complete the planned compression therapy for the first 6 months after liposuction for lymphedema in the lower limb. UEL group, all patients in this group completed the planned compression therapy for the first 6 months after liposuction for lymphedema in the upper limb. Therefore, there is only a UEL compliance group and no UEL non-compliance group. Abbreviations: LEL, lower extremity lymphedema; UEL, upper extremity lymphedema.

**Figure 4 jcm-12-01727-f004:**
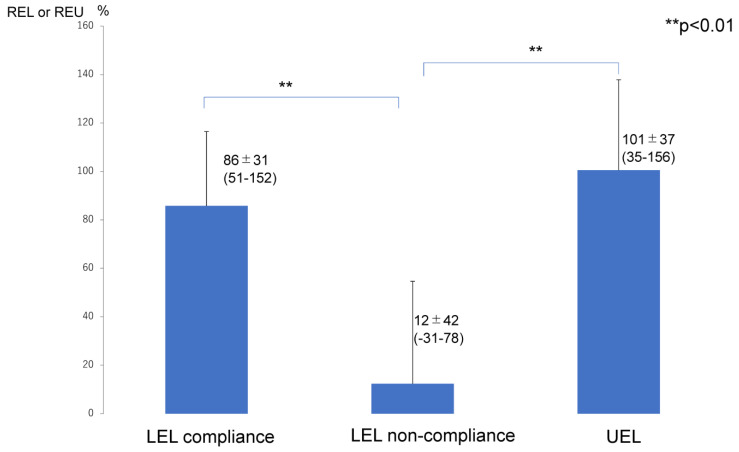
Changes in REL or REU after surgery. LEL compliance group, patients who completed the planned compression therapy for the first 6 months after liposuction for lymphedema in the lower limb. LEL non-compliance group, patients who did not complete the planned compression therapy for the first 6 months after liposuction for lymphedema in the lower limb. UEL group, all patients in this group completed the planned compression therapy for the first 6 months after liposuction for lymphedema in the upper limb. Therefore, there is only a UEL compliance group and no UEL non-compliance group. REL was calculated as ([preoperative PEL] − [postoperative PEL])/[preoperative REL] × 100. REU was calculated as ([preoperative PEU] − [postoperative PEU])/[preoperative REU] × 100. Abbreviations LEL, lower extremity lymphedema; PEL, percentage of excess LEL index in the affected limb; PEU, percentage of excess UEL index in the affected limb; REL, reduction in PEL after surgery; REU, reduction in PEU after surgery; UEL, upper extremity lymphedema.

**Figure 5 jcm-12-01727-f005:**
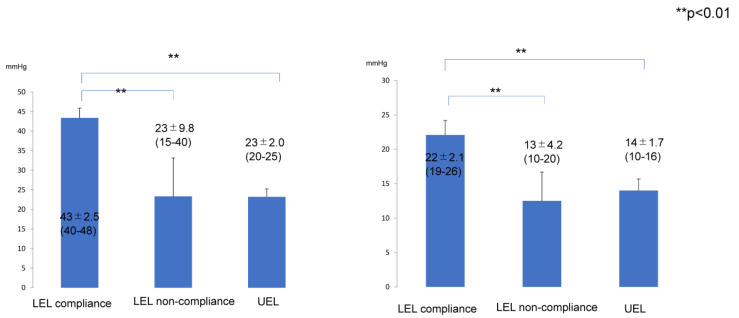
Compression pressure on the leg or forearm (**left**) and on the thigh or arm (**right**) in the LEL and UEL subgroups. LEL compliance group, patients who completed the planned compression therapy for the first 6 months after liposuction for lymphedema in the lower limb. LEL non-compliance group, patients who did not complete the planned compression therapy for the first 6 months after liposuction for lymphedema in the lower limb. UEL group, all patients in this group completed the planned compression therapy for the first 6 months after liposuction for lymphedema in the upper limb. Therefore, there is only a UEL compliance group and no UEL non-compliance group. Abbreviations: LEL, lower extremity lymphedema; UEL, upper extremity lymphedema.

**Figure 6 jcm-12-01727-f006:**
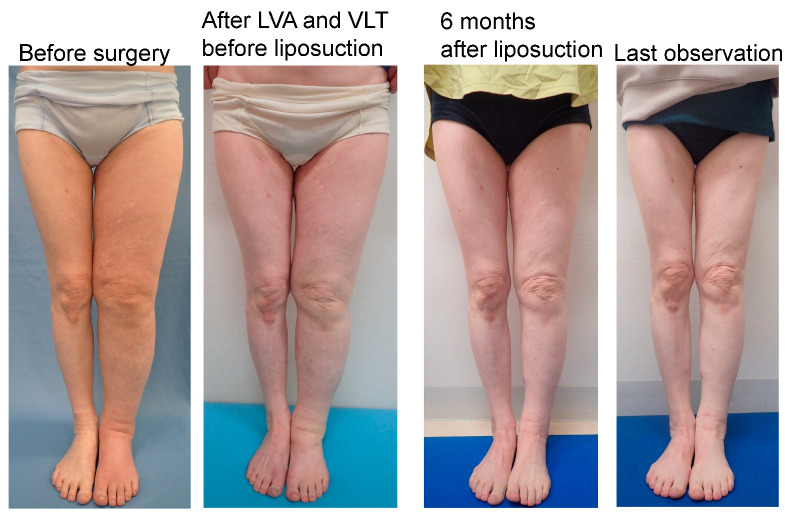
Representative case from the LEL compliance group. The case was a 61-year-old woman with left-sided LEL secondary to ovarian cancer (BMI 18.77, observation period 24 months, 11 LVAs and 2 VLTs in the affected extremity, and a total liposuction volume of 750 mL). The LEL index was 340 before surgery, 332 before liposuction and after LVA and VLT, 240 at 6 months after liposuction, and 244 at the last observation. The rate of improvement in the LEL index was 27.7% at 6 months after surgery and 26.5% at the last observation. The PEL index was 73% before surgery, 71% before liposuction and after LVA and VLT, 24% at 6 months after liposuction, and 26% at the last observation. The REL was 66.7% at 6 months after surgery and 63.8% at the last observation. Abbreviations: BMI, body mass index; LEL index, lower extremity lymphedema index; LVA, lymphovenous anastomosis; PEL, percentage of excess LEL index in the affected limb; REL, reduction rate in PEL; VLT, vascularized lymphatic transfer.

**Figure 7 jcm-12-01727-f007:**
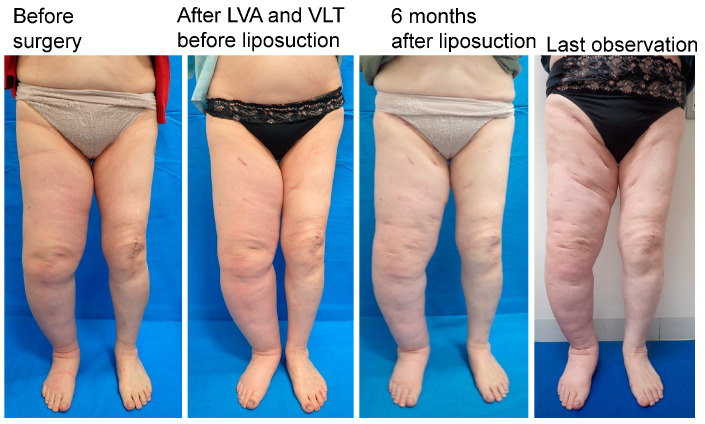
Representative case from the LEL non-compliance group. The patient was a 75-year-old woman with right-sided primary LEL (BMI 30.82, observation period 20 months, 8 LVAs and 2 VLTs in the affected extremity, and a total liposuction volume of 4400 mL). The LEL index was 295 before surgery, 289 before liposuction and after LVA and VLT, 320 at 6 months after liposuction, and 314 at the last observation. The rate of change in the LEL index was −10% at 6 months after surgery and −0.1% at the last observation. The PEL index was 40% before surgery, 38% before liposuction and after LVA and VLT, 53% at 6 months after liposuction, and 50% at the last observation. The REL was −38% at 6 months after surgery and −31% at the last observation. Abbreviations: BMI, body mass index; LEL index, lower extremity lymphedema index; LVA, lymphovenous anastomosis; PEL, percentage of excess LEL index in the affected limb; REL, reduction rate in PEL; VLT, vascularized lymphatic transfer.

**Figure 8 jcm-12-01727-f008:**
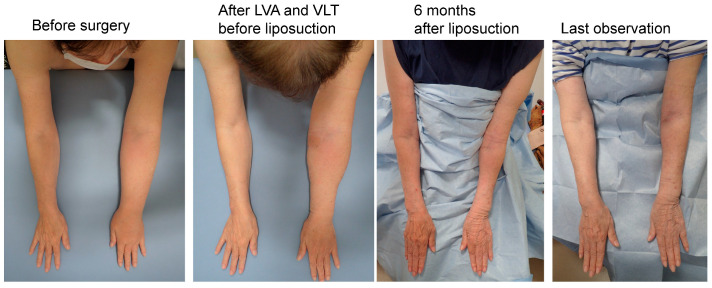
Representative case from the UEL group. The case was a 72-year-old woman with left-sided UEL secondary to breast cancer (BMI 22.72, observation period 24 months, 18 LVAs and 1 VLT in the affected extremity, and a total liposuction volume of 900 mL). The UEL index was 128 before surgery, 125 before liposuction and after LVA and VLT, 72 at 6 months after liposuction, and 76 at the last observation. The rate of improvement in the UEL index was 42.4% at 6 months after surgery and 39.2% at the last observation. The PEU index was 40% before surgery, 34% before liposuction and after LVA and VLT, −22.6% at 6 months after liposuction, and −18.3% at the last observation. The REU was 166% at 6 months after surgery and 153% at the last observation. Abbreviations: BMI, body mass index; UEL, upper extremity lymphedema; LVA, lymphovenous anastomosis; PEU, percentage of excess UEL index in the affected limb; REU, reduction rate in PEU, VLT, vascularized lymphatic transfer.

**Table 1 jcm-12-01727-t001:** Formulas used to evaluate the volume of extremities with lymphedema.

ITEM	Formula
**LEL index:** lower extremity lymphedema index	[total of squares of the circumference at five areas of each lower extremity]/BMI.
**UEL index:** upper extremity lymphedema index	[total of squares of the circumference at five areas of each lower extremity]/BMI.
**Improvement rate in LEL index**	{([preoperative LEL index ] − [postoperative LEL index])/[preoperative LEL index]}× 100
**Improvement rate in UEL index**	{([preoperative UEL index] − [postoperative UEL index])/[preoperative UEL index]}× 100
**PEL:** percentage of excess LEL index for affected lower extremity	{([LEL index of affected extremity] – [LEL index of unaffected contralateral extremity])/[LEL index of unaffected contralateral extremity]} × 100
**PEU:** percentage of excess UEL index for affected upper extremity	{([UEL index of affected extremity] – [UEL index of unaffected contralateral extremity]) [UEL index of unaffected contralateral extremity]} × 100
**REL:** reduction rate in PEL	{([preoperative PEL index ] − [postoperative PEL index ])/[preoperative PEL index]} × 100
**REU:** reduction rate in PEU	{([preoperative PEU index] − [postoperative PEU index])/[preoperative PEU index]} × 100

**Table 2 jcm-12-01727-t002:** Patient demographics and clinical characteristics according to whether liposuction was performed for lower or upper extremity lymphedema.

	Lower Extremity Lymphedema	Upper Extremity Lymphedema	*p*-Value	
**Male:Female (n)**	2:16	1:9	0.9	chi-squared test
**Primary:Secondary (n)**	6:12	3:7	0.9	chi-squared test
**Age (years)**	61.1 ± 16.0 (26–79)	62.4 ± 9.4 (49–73)	0.8	Student’s *t*-test
**Body mass index**	23.7 ± 3.7 (18.8–30.8)	26.8 ± 4.4 (19.2–32.3)	0.06	Student’s *t*-test
**Observation period (months)**	16.7 ± 4.9 (12–25)	18.9 ± 5.3 (12–25)	0.27	Student’s *t*-test
**Average time difference between last physiological reconstruction (LVA or VLT) and liposuction (months)**	15.6 ± 8.8 (8–29)	14.3 ± 5.2(8–25)	0.67	Student’s *t*-test
**LVAs in affected extremity** **(anastomoses, n)**	11 ± 3.6 (6–22)	9.6 ± 3.6 (5–18)	0.3	Student’s *t*-test
**VLTs in affected extremity (** **lymph node transfers, n)**	1.4 ± 0.6 (1–3)	1.5 ± 0.8 (1–3)	0.7	Student’s *t*-test
**LEL or UEL index (before surgery) (%)**	298 ± 41.6 (211–378)	123.7 ± 18.8(103-158)		
**PEL or PEU index (before surgery) (%)**	37.9 ± 20.6 (6.4–74.2)	33.9 ± 10.3 (17.6–49.1)	0.6	Student’s t
**Total liposuction volume (ml)**	2242 ± 1190 (500–4400)	1260 ± 622 (650–2200)	0.02	Student’s t
**LEL or UEL index (after surgery; last observation month) (%)**	248 ± 31.6 (203–314)	92.7 ± 10.9(76–106)		
**Improvement rate (after surgery; last observation month)** **(%)**	16.6 ± 9.3 (−0.1–27.2)	24.3 ± 8.9 (9.8–39.2)	0.04	Student’s t
**PEL or PEU (** **after surgery; last observation month) (%)**	14.9 ± 15.8 (−11.4–50.2)	1.2 ± 13.1 (−18.3–31.4)	0.03	Student’s t
**REL or REU** **(** **after surgery; last observation month) (%)**	59.3 ± 49.4 (−31.8–152)	100.1 ± 37.3 (35.3–155.6)	0.03	Student’s t
**Compression pressure for initial 6 months after liposuction (mmHg)**	**Foot to leg**	**hand to forearm**		
	34.4 ± 9.7 (15–40)	23.2 ± 2.0 (20–25)	0.001	Student’s t
	**Thigh**	**arm**		
	17.5 ± 4.3 (10–20)	14 ± 1.7 (10–16)	0.02	Student’s t
**Compression pressure between 6 months and last observation after liposuction (mmHg)**	**Foot to leg**	**hand to forearm**		
	23.4 ± 4.3 (15–28)	12.1 ± 2.0 (10–15)	0.03 × 10^−6^	Student’s t
	**Thigh**	**arm**		
	10.9 ± 1.1 (9–13)	6.8 ± 1.0 (5–8)	0.04 × 10^−8^	Student’s t
**Implementation status of planned compression for initial 6 months after liposuction**				
**Compliance (cases, n)**	12	10	0.04	chi-square test
**Non-compliance (cases, n)**	6	0		

Postoperative refers to data measured in the last month of the observation period. Abbreviations: BMI, body mass index; LEL index, lower extremity lymphedema index; LVA, lymphovenous anastomosis; PEL, percentage of excess LEL index in the affected limb; PEU, percentage of excess UEL index in the affected limb; UEL index, upper extremity lymphedema index; VLT, vascularized lymphatic transfer.

## Data Availability

The data presented in this study are available on request from the corresponding author S.Y. The data are not publicly available due to privacy restrictions.

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
