# Peer review of "Comparison of the Effectiveness of Liposuction for Lower Limb versus Upper Limb Lymphedema"

_jcm, 2023, doi:10.3390/jcm12051727_

Round 1

Reviewer 1 Report

I would limit lower extremity cases and increase cases for LE.

UE vs LE comparison is like comparing apple to orange.

Author Response

Reviewer 1.

I would limit lower extremity cases and increase cases for LE. UE vs LE comparison is like comparing apple to orange.

Thank you for spending your valuable time to review our article. We are sorry for not being able to obtaine the reviewer’s high estimation with the concept of the study. Next opportunity, we will submit the study limited in LE or UE in which the number of cases is increased.

Reviewer 2 Report

In this study, the authors retrospectively compared the effectiveness of liposuction in our patients according to whether it was performed for LEL or UEL and identified factors with outcomes. The conclusion: Liposuction seems to be more effective in UEL than in
LEL, probably because the compression therapy required for management after liposuction is easier to implement in UEL. The lower pressure and smaller coverage area required for postoperative management after liposuction in the upper limb may explain why liposuction for lymphedema is more effective in in UEL than in LEL.

Overall this is an interesting article. Some comments:

1. This is a retrospecitve study and hence a limitation. Please modify

2. The diagnosis was only lymphedema or sometimes lipedema? Is there a difference between the 2 entities and postoperative management?

3. Which liposuction technique was applied? Power assisted? Water assisted?

4. How about postoperative rehabiliation such as lymphdrainage and pressotherapy?

5. Was lymphdrainage performed before the surgery?

6. What is the average time difference between LVA and liposuction?

7. 6mm Cannula in my opinion for lower extremity is too big, recommended is 3mm at least for cosmetic liposuctions. Please comment.

8. Which technique was used? Tumescent? Super-Wet technique?

Author Response

Reviewer 2.

Thank you for spending your valuable time to review our article.

We accepted the reviewer’s suggestions and revised the article.

  1. This is a retrospecitve study and hence a limitation. Please modify

The authors have added the following sentences as limitations, “This study has some limitations. First, it involved a small number of patients and the longest observation period was 25 months. Therefore, longer-term investigations in a larger sample are necessary. There is also a need for a more accurate assessment method. For example, we assessed compliance with application of compression pressure only by observation and interview at each follow-up visit during the study. A more objective method, such as having patients record daily compliance with application of compression pressure, may produce more accurate results.”.

  1. The diagnosis was only lymphedema or sometimes lipedema? Is there a difference between the 2 entities and postoperative management?

In this study, diffuse patterns were observed by ICG lymphography in affected limbs whole area of all cases. Therefore, we judged all the cases as lymphedema.

The authors do not know the clear definition of difference between lymphedema and lipedema, however, has the impression the lymphatic function in lipedema is not so severer than lymphedema, hence, management of lipedema after liposuction seems easier than severe lymphedema. However, the authors do not have the data prepared to be published.

The issue the reviewer proposed is very interesting, so that the authors would like to investigate it in the next study.

  1. Which liposuction technique was applied? Power assisted? Water assisted?

The authors used a non-assisted liposuction device (Lead S-200, Kakinuma Medical, Inc. Tokyo, Japan). The authors have added the following words “using a non-assisted liposuction device (Lead S-200, Kakinuma Medical, Inc. Tokyo, Japan).”.

  1. How about postoperative rehabiliation such as lymphdrainage and pressotherapy?

Selfcare, such as self-lymph-drainage, was instructed at the first visit to our outpatient clinic, however, additional rehabilitation such as lymph-drainage was not performed except for the instruction about wearing the compression garments before and after liposuction.

The authors have added the following sentences at the end of “Postoperative management”, “Instructions on self-care, such as self-lymph drainage, were provided at the first visit to our outpatient clinic; however, additional rehabilitation, such as lymph drainage, was not performed except for the instruction about wearing the compression garments before and after liposuction.”.

  1. Was lymphdrainage performed before the surgery?

Selfcare, such as self-lymph-drainage, was instructed at the first visit to our outpatient clinic, however, additional rehabilitation such as lymph-drainage was not performed except for the instruction about wearing the compression garments before and after liposuction. The authors have added the following sentences at the end of “Postoperative management”, “Instructions on self-care, such as self-lymph drainage, were provided at the first visit to our outpatient clinic; however, additional rehabilitation, such as lymph drainage, was not performed except for the instruction about wearing the compression garments before and after liposuction.”.

  1. What is the average time difference between LVA and liposuction?

The average time difference between last physiological reconstruction (LVA or VLT) and liposuction was 15.6±8.8 (8-29) months in LEL and 14.3±5.2(8-25) months in UEL, p value was 0.67. The authors have added these data in Table 1.

  1. 6mm Cannula in my opinion for lower extremity is too big, recommended is 3mm at least for cosmetic liposuctions. Please comment.

The authors have added the following sentences before the limitations at the end of discussion, “Given that volume reduction was prioritized in our study, a 6-mm cannula was used for liposuction; however, a 3-mm cannula is recommended from the viewpoint of cosmesis.”.

  1. Which technique was used? Tumescent? Super-Wet technique?

The authors used Tumescent technique. It was added.

Reviewer 3 Report

1. it would be better if a separate table was given regarding the formula included in the method

2. please be consistent in using the numbers listed in the results

Author Response

Reviewer 3.

Thank you for spending your valuable time to review our article.

We accepted the reviewer’s suggestions and revised the article.

  1. it would be better if a separate table was given regarding the formula included in the method

The authors made a separate table regarding the formula for the items to evaluate the lymphedema as Table 1.

  1. please be consistent in using the numbers listed in the results.

The authors checked the numbers and lusted in the results.
